# Tissue Engineering as a Promising Treatment for Glottic Insufficiency: A Review on Biomolecules and Cell-Laden Hydrogel

**DOI:** 10.3390/biomedicines10123082

**Published:** 2022-11-30

**Authors:** Wan-Chiew Ng, Yogeswaran Lokanathan, Marina Mat Baki, Mh Busra Fauzi, Ani Amelia Zainuddin, Mawaddah Azman

**Affiliations:** 1Department of Otorhinolaryngology-Head and Neck Surgery, Faculty of Medicine, Universiti Kebangsaan Malaysia, Kuala Lumpur 56000, Malaysia; 2Centre for Tissue Engineering and Regenerative Medicine, Faculty of Medicine, Universiti Kebangsaan Malaysia, Kuala Lumpur 56000, Malaysia; 3Department of Obstetrics and Gynaecology, Faculty of Medicine, Universiti Kebangsaan Malaysia, Kuala Lumpur 56000, Malaysia

**Keywords:** glottic insufficiency, vocal fold injection, incorporation, cell-laden, biomolecule

## Abstract

Glottic insufficiency is widespread in the elderly population and occurs as a result of secondary damage or systemic disease. Tissue engineering is a viable treatment for glottic insufficiency since it aims to restore damaged nerve tissue and revitalize aging muscle. After injection into the biological system, injectable biomaterial delivers cost- and time-effectiveness while acting as a protective shield for cells and biomolecules. This article focuses on injectable biomaterials that transport cells and biomolecules in regenerated tissue, particularly adipose, muscle, and nerve tissue. We propose Wharton’s Jelly mesenchymal stem cells (WJMSCs), induced pluripotent stem cells (IP-SCs), basic fibroblast growth factor (bFGF), vascular endothelial growth factor (VEGF), hepatocyte growth factor (HGF), insulin growth factor-1 (IGF-1) and extracellular vesicle (EV) as potential cells and macromolecules to be included into biomaterials, with some particular testing to support them as a promising translational medicine for vocal fold regeneration.

## 1. Introduction

Voice disorders affect 16.9% of the adult population (aged 18 and more) and 13.1% more of the elderly population (aged 85 and more) [1,2,3]. Glottic insufficiency is diagnosed when the vocal fold does not entirely close during phonation [4]. It impairs voice production and the protection of the lower airway, resulting in impaired social function, decreased work performance, and an increased risk of aspiration. Due to the unique characteristics of the human vocal fold and the numerous causes of glottic insufficiency, it is difficult to recommend the optimal treatment for this illness. Tissue engineering has advantages in this field since it strives to enhance regeneration and provides longer-lasting or even permanent vocal fold augmentation [5,6]. Tissue engineering has been researched extensively in several regenerative techniques, including cartilage, neuron, cardiac, and bone regeneration [7,8,9,10]. Nonetheless, previous studies [11,12,13,14] have identified vocal fold fibroblasts, muscle progenitor cells, embryonic stem cells (ESCs), bone marrow mesenchymal stem cells (BMMSCs), and adipose stem cells (ASCs) with or without the use of a scaffold as a delivery vehicle for vocal fold regeneration. This study seeks to outline the most recent advancements in injectable biomaterials that transport biomolecules and cells for regeneration purposes and to identify future directions for tissue engineering–based treatment of glottic insufficiency.

### 1.1. Structure of Vocal Fold

Three layers—cover, transition, and body—are thought to make up a human vocal fold [15]. Collagen, elastin, hyaluronic acid (HA), decorin, and fibronectin make up the majority of the extracellular matrix (ECM) proteins found in the lamina propria [16]. The superficial layers of lamina propria and epithelium in the cover have vibratory qualities which are crucial for phonation. Superficial lamina propria comprises loosely packed connective tissue [17]. The transition layer is made up of an intermediate layer primarily of elastin and deep layer of collagen. Collagen provides mechanical support to the vocal fold while elastin maintains the elasticity of the vocal fold [18,19]. The vocalis muscle makes up the body layer, which forms the base of this intricate three-dimensional structure [15]. Interestingly, cell junctions hold stratified squamous epithelium, which serves as a protective layer around membranous vocal folds [20]. Compared to newborn epithelial cells, adult epithelial cells displayed more significant intercellular gaps, greater mechanical strength, and more excellent elasticity [21]. In the lamina propria, the vascular network disperses differently. Only capillaries are seen in the superficial lamina propria; arterioles and venules are located in the intermediate and deep lamina propria. In muscular tissue, bigger vessels are more prevalent. Pericytes have been seen on capillaries, and it is thought that pericytes can shield capillaries from lamina propria vibration. Additionally, pericytes are found to be crucial for angiogenesis [22].

The paraglottic area, which houses intrinsic laryngeal muscle, nerve connections, and adipose tissue, is connected laterally to the vocal fold [23]. Intrinsic laryngeal muscles include thyroarytenoid (TA), lateral-cricoarytenoid (LCA) and interarytenoid (IA) muscles for adduction, and the posterior cricoarytenoid (PCA) muscle for abduction [24]. It is suggested that the TA, cricothyroid (CT), and LCA function as a single muscle stimulated by a motor unit [25]. Recurrent laryngeal nerves (RLN) and superior laryngeal nerves (SLN) provide intrinsic and extrinsic impulses, respectively [26]. RLN perform adduction and abduction functions [27]. The inability to change vocal pitch due to CT motor loss is directly related to SLN dysfunction [28]. The RLN innervates the thyroarytenoid-lateral cricoarytenoid (TA-LCA) for adduction, whereas the SLN innervates the CT. It can be summarized that the composition of the vocal fold includes elastic components which enable its phonation properties, mechanically strong components to support its structure, blood capillaries to provide nutrients to cells, and nerves to control its movement. The structure of the human vocal fold is briefly demonstrated in Figure 1.

### 1.2. Etiologies of Glottic Insufficiency

It is vital for the clinician to differentiate the causes of glottic insufficiency in opting for suitable treatments. The main symptoms of glottic insufficiency include being unable to generate an effective voice and being unable to protect the lower airway during swallowing. The most common cause of glottic insufficiency is vocal fold paralysis/paresis. Simply put, dysfunctional nerves or muscles are the primary causes of the vocal fold’s inadequate closure. Figure 2 explains the physiological characteristics of the normal condition and glottic insufficiency.

Vocal fold paralysis/paresis is identified when the RLN or SLN are damaged, causing the inability of the intrinsic laryngeal muscles to contract [24,29,30], contributing to the inability to move the vocal fold. Vocal fold paresis is defined when the nerve is partially damaged, causing incomplete signaling or abnormal signaling of nerve; paralysis is diagnosed when the vocal fold is not able to move completely [26]. Vocal fold paresis/paralysis has many etiologies, including scarring, iatrogenic disorders, malignancy, central nervous system pathology, and systemic illnesses [31]. An idiopathic cause is searched for when an aetiology cannot be determined after a comprehensive study.

Vocal fold atrophy is characterized when there is dissipation of muscle and loss of intonation even though the TA-LCA complex is mobile within a certain range [32]. Presbyphonia, child/adolescent, and inborn vocal fold scar are the three distinct types of vocal fold atrophy [33,34,35,36]. Commonly, the decreased sensitivity or malfunction of the contractile components inside the TA muscle is linked to the pathophysiology of vocal fold atrophy [1]. One of the possible causes of an ageing voice is structural changes in the vocal fold’s lamina propria [37]. The lamina propria became stiffer from increased collagen density and decreased elastin and HA density. Additionally, it was discovered that the activity of collagenase decreased in ageing vocal folds [16,38].

## 2. Current Treatment for Glottic Insufficiency and Limitations

### 2.1. Surgery and Complementary Treatment

Traditional intervention for glottic insufficiency involves improving glottic closure. As such, type 1 medialization thyroplasty is one of the treatments for glottic insufficiency. It involves implantation of a foreign material into the paraglottic space. It is durable but has some drawbacks, such as a difficult technique and the possibility of vocal fold vibratory function loss over time [39]. Gore-Tex ribbon (polytetrafluoroethylene) and Silastic (silicone rubber) are examples of inert materials used in this surgical procedure [40]. Nonselective laryngeal reinnervation (NSLR) is performed by connecting the ansa cervicalis to the RLN. In observations lasting up to 12 months, NSLR has been proven to improve the voice result of patients considerably [41]. However, NSLR is a technically challenging operation requiring a lengthy process performed under general anesthesia. In patients with bilateral vocal fold immobility, treatment focuses on improving respiration rather than the voice where carbon dioxide laser posterior cordectomy has been described with favorable outcomes [42].

Voice therapy is also widely suggested by practitioners. Previous research assessed voice therapy’s efficacy in RLN and SLN activation in canine vocal fold paresis and paralysis [29]. It was determined that phonation is superior during RLN activation but contracting during SLN activation. Voice therapy and surgical intervention were examined in a separate clinical investigation, demonstrating positive effects on phonation for both arms. However, surgery was performed on patients with more severe symptoms [43].

### 2.2. In-Office Injection: Injectate Type

Vocal fold injection is less expensive and time-consuming than other procedures. Among materials that have been used for injection are paraffin, Teflon, autologous fat, bovine collagen, carboxymethylcellulose (CMC), calcium hydroxyapatite (CaHA) and HA [44,45]. CaHA could withstand more prolonged periods of augmentation than CMC, and practitioners stated that CaHA injections required more force. Despite their distinct qualities, the voice results are comparable [46].

Fat injection was examined as a treatment for individuals with unilateral vocal fold paralysis (UVFP) and vocal atrophy/scar [47]. This study concluded that fat augmentation is suitable for UVFP since it is autologous and durable but not for vocal fold atrophy/scar because it reduces the vocal range. Nonetheless, fat augmentation in conjunction with platelet-rich plasma (PRP) reduced the recovery period and allergic response in type II sulcus vocalis compared to fat augmentation alone [48]. In adipose tissue engineering, autologous fat was injected into vocal muscle and paraglottic spaces to enhance neovascularization and prevent bulk injection at one site [49]. PRP is rich in various growth factors that can assist in regenerating local tissue and provide anti-inflammatory properties [50,51]. However, direction injection of PRP is not sustainable over time, and it requires re-injection to obtain a sustainable outcome.

## 3. Tissue Engineering as a Promising Treatment for Glottic Insufficiency

### 3.1. Tissue Engineering in Vocal Fold Injection

Regenerative medicine is the approach of reinstating human cells, tissue or organs to their usual role [52]. Tissue engineering is application of biomaterial with or without cell transplantation to encourage endogenous regeneration and regain functional tissues or organs [53]. Fillers such as Teflon, polydimethysilicone and calcium hydroxyapatite are commonly injected into the vocal folds to improve glottal closure but are linked with the risk of inflammation, migration, and granuloma development [54]. Moreover, current clinical trial research for vocal fold injection focuses primarily on biomaterial alone (ClinicalTrials.gov number: NCT04700566, NCT03790956, NCT02163772) or direct injection of biomolecules or cells (ClinicalTrials.gov number: NCT05354544, NCT05385159, NCT03749863, NCT02622464, NCT02120781, NCT02904824, NCT04839276). It is suggested that combining biomaterials with biomolecules such as growth factors can improve the efficacy [55]. The combination of biomolecules, cells, and a scaffold serves as a unique delivery system. As cells proliferate to generate new tissue, biomolecules promote the growth of new tissue, and the scaffold serves as an environment for the regeneration of new tissue [56,57]. The comparison between injection of cells and biomolecules without and with a scaffold is shown in Figure 3.

Metals, ceramics, and polymers are the three most common types of scaffold biomaterials [58]. Polymeric hydrogel is often utilized for injecting vocal folds due to its ease of usage, biomimicry, and capacity to produce irregular shapes in confined spaces [59]. Polymeric hydrogel can be fabricated using synthetic or natural molecules [60], as shown in Appendix A. It can be crosslinked by either physical or chemical processes. As a result, chemically crosslinked hydrogels exhibited a rapid gelation reaction and may be suited for injection into the vocal folds [61,62]. Besides chemical crosslinking, this study utilized click chemistry to generate a hydrogel capable of reducing the glottal gap in a rabbit model [63]. Thermosensitive hydrogel is also suitable for vocal fold injection since it solidifies at body temperature [64]. Direct injection of cells is a concern, as this delivery method causes ineffective cell survival and retention [65]. Figure 4 demonstrates that integrating cells into hydrogel reduces this issue and prolongs biomolecules’ transport, enhancing the regeneration impact in scarred vocal folds. Hydrogel is excellent for medication delivery due to its longer-lasting release and environmental sensitivity [66]. Hydrogel has a high water content and mimics the milieu of living organisms [67]. Cell encapsulation can be performed either during or after scaffold creation, provided that the process does not harm the cells [68].

Another benefit of the hydrogel as a delivery vehicle is the precise control of its re-release period. The hydrogel can be engineered to degrade at a rate that is commensurate with the time required for tissue regeneration [69]. Cell fate in the hydrogel is closely associated with degradation rate and substrate modification. As hydrogel degrades, stem cells differentiate into mature parent cells such as chondrocytes and osteocytes [67]. Moreover, a biodegradable gelatin hydrogel microsphere was able to progressively release bFGF in a rabbit model with wounded vocal folds [70]. Nevertheless, natural polymeric hydrogel is subject to fast biodegradation due to its low mechanical properties. The mechanical properties of the hydrogel can be finetuned via different crosslinking strategies such as click chemistry, interpenetrating networks, nanocomposites and more [71]. During finetuning of hydrogel, the microenvironment changes attributable to the crosslinking mechanism will impact the reactions of macrophages and fibroblasts [72]. It is very important to develop suitable mechanical properties of hydrogel, which has similar stiffness to that in body tissue and will help to regulate native fibroblast expression [73]. The interaction between an encapsulated cell and hydrogel is insufficiently evidenced; however, cell encapsulation in the hydrogel was able to modify the breakdown rate of hydrogel, as this study demonstrated that hydrogel containing Schwann cells degraded quicker than hydrogel alone [74]. Cell-hydrogel interactions, such as cell adhesion and movement, must be elucidated to prove the viability of hydrogel applications [75]. Cell adhesion in the hydrogel guarantees its capacity to integrate with native tissue throughout the healing process. Having a connection with this, cell adhesion receptors include the tripeptide arginine-glycine-aspartic acid (RGD) by facilitating cell communication via integrin dimers. Other receptors that stimulate cell adhesion include ligands derived from laminin (YIGSR) and collagen (GFOGER) [76]. Cell-hydrogel interaction is also affected by the hydrogel’s physicomechanical properties, such as its rigidity, composition, viscoelasticity, and microenvironment [76,77]. 

Numerous preliminary studies of hydrogel use for various regenerative reasons have been done. Examples include neurons, wound, tendons, bone, and muscle [78,79,80,81,82,83]. In a 3D hydrogel, Schwann cells express laminin and collagen IV, which may enhance axonal development [74]. A literature research was undertaken to outline the types of cells and macromolecules used to encapsulate in an injectable hydrogel, as shown in Appendix A, and to indicate their possible application in vocal fold regeneration. The subject is separated into two subsections: cell encapsulation and biomolecule encapsulation. Combinations of the keywords “hydrogel”, “stem cell”, “growth factor”, “secretome”, “fibroblast”, “cell”, “incorporation”, “encapsulation”, “delivery”, and “cell-laden” were used to search Web of Science (WoS) for relevant publications. There were a total of 17,799 articles found, and irrelevant articles were omitted; 151 articles on injectable biomaterials for biomolecule and cell encapsulation in adipose, angiogenesis, muscle, and nerve regenerative medicine were included. 

### 3.2. Injectable Hydrogel as Cell Delivery Vehicle

The majority of research develops encapsulation techniques for broad regeneration objectives. Muscle regeneration is followed by angiogenesis, nerve regeneration, and adipose tissue engineering in terms of the number of relevant studies. Comparatively, more research tried to encapsulate cells alone in biomaterials, followed by encapsulation of growth factor, MSC-extracellular vehicle (EV), and siRNA (siRNA). As illustrated in Appendix A, three major categories of cells were examined.

In clinical studies, direct injection of adipose-derived stem cells (ASCs) improved voice outcomes in patients with vocal fold scarring and glottic insufficiency [84,85]. Numerous in vivo investigations involving direct injection of ASCs have shown that ASCs can upregulate HA while downregulating collagen type I, type III, matrix metalloproteinase (Mmp1), and Mmp8 expression [86,87,88]. ASCs were also intimately linked to the secretion of FGF2, HGF, and basic fibroblast growth factor (bFGF). However, with direct injection, ASCs were only able to survive for 14 days; encapsulation helps to circumvent this problem [89,90]. It is uncertain whether ASCs or BMMSCs are more effective for augmenting vocal folds. Hiwatashi and colleagues recommended ASCs because they would increase HA control more effectively than BMMSCs [91]. Bartlett and colleagues recommended otherwise [92]. By stimulating BMMSCs with transforming growth factor beta (TGF-β), differentiation of vocal fold fibroblast into myofibroblast is inhibited [93]. Few investigations demonstrated that the qualities of ASCs are superior to those of BMMSCs because they are more stable, anti-inflammatory, proliferative, and have the same capacity to differentiate into various lineages [94,95,96,97]. The anti-fibrosis function of ASCs is depicted in Figure 5.

Human umbilical cord-mesenchymal stem cells are also known as Wharton’s Jelly-mesenchymal stem cells (WJMSCs). A WJMSC is a mucous connective tissue in the umbilical cord, which is usually discarded [98]. It is able to differentiate into different types of cells, namely mesoderm (adipocyte, osteocyte, chondrocyte), ectoderm (nerve) and endoderm (islet and liver cells) [99]. The addition of nerve growth factor (NGF) to WJMSCs in a collagen scaffold has been shown to enhance the differentiation and healing of a wounded RLN in rabbits [100]. In addition, WJMSCs have low immunogenicity with respect to T cells, B cells, dendritic cells, natural killer (NK) cells, neutrophils, and mast cells [101]. Insufficient evidence supports the use of WJMSCs in vocal fold augmentation. As it can potentially regenerate myocyte [102,103], nerve cell [104,105,106] and adipocytes, which are abundant in vocal folds, it has a strong potential to repair atrophied or damaged tissue in the vocal fold. With its superior immunomodulatory capabilities, it can prevent inflammation and minimise vocal fold fibrosis. Figure 6 depicts the role of WJMSCs in neuron and muscle regeneration, while Figure 7 summarises their immunomodulatory features. 

iPSCs are a relatively novel use for augmenting vocal folds. Previous research indicated its ability to rebuild injured muscle in rat model vocal folds [107]. iPSCs have the potential to develop into epithelial cells and alleviate fibrosis when incorporated into hydrogel [108,109]. iPSCs have been shown to develop into endothelial cells when exposed to growth factors such as activin A, bone morphogenetic protein 4 (BMP4), and bFGF. However, the application of iPSCs in vocal fold augmentation has yielded limited results. Additionally, it can repair skeletal muscle and Schwann cells [110,111]. However, it is known that iPSCs are tumorigenic and carry a high risk for clinical application [112]. In order to resolve this problem, additional investigation and comprehension of its underlying mechanics are required. 

Native tissue serves a crucial role in facilitating wound healing. Myofibroblast stimulation from fibroblast is known to enhance the creation of ECM proteins. TGF-1 facilitates the procedure [113]. Epithelial cells, fibroblasts, macrophages, and platelets produce TGF-β. It has activities in both promoting and inhibiting wound healing; for example, it inhibits airway epithelial growth but promotes mucosal remodelling. The vocal fold wound healing process is regulated by epidermal growth factor (EGF) and transforming growth factor beta 1 (TGF-1) [114]. Following injury, epithelial cells increased in thickness and permeability [115]. Branco and his colleague discovered that the lamina propria and epithelial layers of ageing vocal folds tend to atrophy [116]. Therefore, the structural alterations of the vocal fold’s native tissue are essential for maintaining its efficient vibrational state.

### 3.3. Injectable Hydrogel as Biomolecule Delivery Vehicle

As hydrogel has a special affinity for water, it can be used as a hydrophilic growth factor delivery system. Based on the application (slow and extended or fast and short release), protein retention and delivery can be modified [117]. A hydrogel with low crosslinking, small particle size and susceptibility to enzymatic degradation will result in quicker growth factor release [118]. Growth factor incorporation will increase the bioactivity of the hydrogel [119]. For instance, Walters and colleagues demonstrated [120] that combining platelet-derived growth factor AB (PDGF-AB) and TGF-1 in collagen hydrogel promotes the differentiation of ASCs into smooth muscle cells. The thermosensitive heparin-poloxamer hydrogel containing bFGF and NGF enhances Schwann cell proliferation via the PI3K/Akt, JAK/STAT and MAPK/ERK signalling pathways [121]. To provide multiple regenerative aims, three different types of growth factors, namely VEGF, PDGF and BMP2, were released from a collagen hydrogel over a 28-day period in order to stimulate angiogenesis in a rat model [122].

As described in Appendix A, the biomolecules employed in biomaterials can be categorised into four categories: neurotrophic growth factor, growth factor, proteins, and extracellular vesicles. In a rabbit model, a collagen scaffold containing NGF and human umbilical MSC was administered. After eight weeks, a positive response was obtained in the RLN injury model [100]. In vocal fold regeneration, brain-derived neurotrophic factor (BDNF), ciliary neurotrophic factor (CNF), and stromal cell-derived factor-1 (SDF-1) have not been researched. The SDF-1/CXCR4-mediated FAK/PI3K/Akt pathway [123] is thought to protect neuron tissue by avoiding cell death and inflammation. Following nerve damage, BDNF and SDF-1 increase in order to repair and modulate cells [124]. Ciliary neurotrophic factor enhances axon regeneration by binding to ciliary neurotrophic factor receptor α and then activating STAT3 [125]. Glottic insufficiency may have various causes, including muscle atrophy, nerve degeneration or damage, and anatomical alterations to the lamina propria. Consequently, treatment with just nerve growth factor may not yield optimal results. 

As the vocal fold is composed of epithelium and fibroblasts, EGF can reconstitute functional mucosa by promoting epithelium regeneration and HA synthesis. Several clinical trials [126,127,128] have demonstrated that direct injection of bFGF into the vocal fold has a beneficial effect on functional voice results. The most effective therapeutic impact on vocal fold atrophy was produced by this treatment. As the majority of instances were caused by ageing, bFGF use was shown to increase fibroblast synthesis of HA, hence lowering collagen deposition It was believed that bFGF treatment was more effective than biomaterial implantation because it altered the vibratory characteristics of the vocal fold and increased its volume [129]. bFGF was not only able to restore the flexibility of the vocal fold but also increased the density of the thyroarytenoid muscle in aged vocal folds [130]. 

Hepatocyte growth factor (HGF) has anti-fibrotic and angiogenesis properties and is produced by mesenchymal cells such as fibroblasts, macrophages, renal mesangium, etc. A clinical investigation demonstrated the efficacy and regeneration potential of HGF in patients with vocal fold scarring and sulcus. By repeatedly injecting the vocal fold with HGF, the patients exhibited a considerably improved outcome while maintaining a high level of safety. An earlier pre-clinical investigation demonstrated that HGF could boost fibroblast synthesis of HA and decrease collagen deposition [131]. In a second in-vitro study of a damaged vocal fold in rabbits, HGF delivered in a hyaluronic/alginate hydrogel was more effective than HGF injection alone [132]. With immediate treatment of HGF after vocal fold injury, collagen synthesis was decreased and angiogenesis was stimulated [133].

There is paucity of data on the effect of PDGF-BB on vocal fold regeneration. One study, however, showed that PDGF-BB can stimulate vessel development in rat models [134]. Similar to PDGF-BB, angiogenin has not been researched in the regeneration of vocal folds. By blocking the TGF-1/Smad pathway, it has been shown to alter fibroblast scar formation [135]. VEGF stimulates the development of blood vessels, which is essential for tissue regeneration. Then, numerous investigations on VEGF in various areas, including skeletal, peripheral nerve, dental pulp, and heart regeneration, were conducted [136,137,138,139,140]. VEGF promotes the development of HUVEC and neurite cells through the Erk/Akt pathway [141]. IGF-1 is one of the key proteins that regulate skeletal muscle metabolism pathways such as PI3K/Akt/mTOR and PI3K/Akt/GSK3β. IGF-1 suppresses cytokines that produce muscle atrophy and myostatin via these signalling pathways by suppressing nuclear factor-kappa beta (NF-ĸB) and Smad pathways. It can also promote skeletal muscle stem cells for the regeneration of skeletal muscle [142]. Insulin growth factor-1 (IGF-1) was induced by myokine/cytokine Meteorin-like that promotes myogenesis [143].

Researchers are currently interested in the exosome from umbilical cord MSC, since it can prevent tumorigenic concerns. It has great promise for use in a variety of conditions, including wounds, type 2 diabetes, inflammatory bowel disease, Alzheimer’s disease, spinal cord injury, myocardial ischemia injury, and graft-versus-host disease [144]. Extracellular vesicles (EVs) can be subdivided into numerous categories. Microvesicles and exosomes are among the subgroups. Microvesicles transport phosphatidylserine-containing proteins, mRNAs, miRNAs, and lipids, while exosomes transport DNA, lipids, RNAs, and proteins [145]. Human umbilical cord EVs have been demonstrated to exert anti-inflammatory qualities (interleukin-10, IL-10) and reduced pro-inflammatory responses (IL-1β, IL-6), followed by improved motor, axon, and Schwann cell regeneration [146]. Myoblasts release EVs containing HGF, IGF-1, TGF-3, VEGF, fibroblast growth factor-β3 (TGF-β3) and fibroblast growth factor 2 (FGF2). These secretions promote cell interaction between myoblasts for proliferation [147]. EVs enhance cross-talk in skeletal muscle to induce glucose release and adipose dissociation [148]. Interestingly, EVs that are secreted by astrocyte support a variety of tasks that range from delivering angiogenic factors such as FGF2, and VEGF to heat shock proteins, synapsin 1 and apolipoprotein-D, which aid in neuronal protection in hazardous settings [149]. EVs play a role in neuronal cell communication and in response to external stimuli such as inflammatory responses and the central nervous system [150]. Although EV can be created from a variety of cell sources, EV derived from MSC is persuasive in its ability to regenerate the central nervous system [151]. Figure 8 summarises the regenerative properties of EVs in neuronal and muscular tissue.

Recent studies [152] on patients with vocal fold atrophy, scarring, and sulcus have examined PRP. PRP is a combination of autologous growth factors that stimulate EGFR secretion to promote wound healing. It includes PDGF, TGF-β, VEGF, EGF and IGF [153]. However, this treatment demonstrated a short augmentation period, which is typically between three to six months [152]. PRP has been demonstrated to increase ECM remodelling and decrease collagen density during vocal fold wound healing [153]. Platelets release cytokines and growth factors that stimulate cell proliferation, angiogenesis, and cell migration during vocal fold injury [154].

Gene therapy aims to transmit genetic material, such as deoxyribonucleic acid (DNA) or ribonucleic acid (RNA), in order to modify the genetic outcome [155]. Nonetheless, gene therapy has created safety concerns, as it may generate negative side effects such as inflammatory reactions [156]. A study sought to increase the effectiveness of silencing prolyl hydroxylase domain-containing protein 2 (PHD2) by delivering siRNA through biodegradable biomaterial, to regulate the inflammatory response of the host and promote angiogenesis upon implantation of biomaterials [157]. PHD2 is one of the essential genes that affect endothelial cell inflammatory properties. By silencing PHD2, it is possible to reduce over-inflammation, which has applications in acute inflammatory diseases [158]. By incorporating biomaterial application into gene therapy, the biomaterial can delay the release of gene materials through progressive degradation and reduce the risk of inflammation [159,160]. Figure 8 indicates the overall regenerative capacity of biomolecules in the vocal fold. 

## 4. Future Study and Limitation

Previous research has proposed two types of cell encapsulation, including stem cells and native cells. The majority of previous research demonstrated the role of native cells in maintaining the vibratory function of vocal folds but not their regenerative capacity. Therefore, the current review suggests that encapsulating stem cells in biomaterials is a more promising technique for augmenting vocal folds. This is because glottic insufficiency may continue, necessitating the use of regenerating biomaterials. Moreover, immunomodulatory characteristics of MSCs can decrease fibrosis during injury [101,161]. In a clinical experiment, cell treatment for vocal fold regeneration utilizing autologous BMMSCs yielded promising results; however, other MSCs sources should be examined to minimize invasive cell harvesting techniques [162]. Future research should focus on incorporating stem cells into biomaterials, particularly WJMSCs and iPSCs, to improve regeneration outcomes. Although WJMSCs and iPSCs have been explored in a variety of applications, including in neuronal tissue, cardiac tissue, skin, cartilage, muscle, and bone, there are few preliminary studies that indicate their potential for vocal fold regeneration [163]. Although both ESCs and iPSCs exhibited the ability to differentiate, the cultivation of ESCs involved ethical problems and the possibility of host rejection, whereas the cultivation of iPSCs needed costly technology and a metagenesis risk [164]. Moreover, some nations restricted the use of ESCs owing to religious practices [165]. Previous research on the application of MSCs to vocal fold regeneration focused exclusively on BMMSCs and ASCs [166]. BMMSCs were the most extensively investigated adult stem cells, but practical translation remained difficult, and ASCs were favored because of their superior stability and differentiation capacity [167]. WJMSCs were found to have superior cell proliferation and immunophenotypic indicators compared to ASCs [168]. Future research must give empirical data on WJMSCs and iPSCs in vocal fold regeneration as a basis for translational research. Future research should examine the efficacy of restoring native vocal folds by direct injection of WJMSCs and with hydrogel, the concentration of WJMSCs, and the sustainability of the treatment. Additional research is required to achieve genomic stability, low tumorigenesis, low toxicity, and low immunogenicity in iPSCs for use in vocal fold regeneration [169]. Future research must also address [170] the type of cell sources to be induced, the method of gene change, and the efficiency of induction.

As stated previously, the lamina propria capillaries are lined with pericytes, and research has shown that pericytes can differentiate from human pluripotent stem cells and are involved in fibrogenesis, angiogenesis, immunomodulation, and differentiation [171,172]. Pericyte-like differentiated ASCs have been found to promote endothelial cells, which aid in retinal vascularization [173]. Although previous studies suggested that pericytes were multipotent and capable of regenerating damaged tissue and promoting healing, a single study [174] contradicted this notion. Therefore, the interaction between injected stem cells and native pericyte is a topic worthy of investigation, since it may be one of the mechanisms underlying angiogenesis.

bFGF and HGF have been clinically studied, and their efficacy has been established [126,127,128,131]. However, the direct injection administration route was utilized. There are currently studies incorporating bFGF into biomaterials that indicate a positive in vivo outcome [70,175]. Future study should determine whether the integration of these two growth factors into biomaterials is effective in prolonging the release period and enhancing regeneration in a shorter time frame. As vocal folds are composed of extensive muscle, nerve, and adipose tissue, this review also proposes that VEGF and IGF-1 may have potential applications in vocal fold regeneration, as VEGF increases the proliferation of epithelial and neuronal cells and IGF-1 promotes muscle regeneration. However, research should investigate the regenerative efficiency, structural changes following administration, and the VEGF and IGF-1 pathways implicated in the vocal fold.

Previous research has suggested that the MSC’s regenerative properties may be due to the release of EVs [176]. EV delivery provides cell-free therapy, eliminating drawbacks such as carcinogenesis, graft-versus-host disease, and instability due to storage and senescence [177,178]. It is debatable in multiple applications, including nervous, cardiac, bone, cartilage, kidney, liver, muscle, and wound healing, but not vocal fold regeneration [179,180]. With that, this review suggests that EVs are a potential biomolecule for use in vocal fold regeneration. As a result of its anti-immunogenic and regenerative properties, it is capable of restoring nerve and muscle function in the vocal fold, as nerve and muscle degeneration or dysfunction are the most common causes of vocal fold paralysis. There are studies in using different sources of EVs, from BMMSCs, WJMSCs or ASCs and more, and the claim for their effectiveness was affirmative. However, the efficacy of EVs derived from different types of cell sources, as well as the comparison between direct injection of EVs and incorporation in an injectable hydrogel, should be elucidated. Nevertheless, isolation procedures, storage conditions, and injection concentrations need to be fully optimized. Only with sufficient pre-clinical data can the clinical application of EVs for individuals with glottic insufficiency be translated. 

Moreover, this review also suggests that future study should explore incorporation of cells and biomolecules together in hydrogel, to obtain better results. The synergetic effect of cells and biomolecules will provide better regenerative outcomes for the native tissue. In short, current tissue engineering in glottic insufficiency has received inadequate study, whereby current progress focuses on direct injection of cells or biomolecules and encapsulation of cells or biomolecules alone; the sources of cells might not be relevant for expansion in clinical studies (for example, the use of BMMSCs). This review proposes that future studies should look into encapsulation of cells and biomolecules together in hydrogel and application of relevant cell sources such as WJMSCs and iPSCs. Table 1 and Table 2 provide a summary of current research on the injection of biomolecules or cells into the vocal folds for regenerative purposes, as well as a list of cells or biomolecules with the potential for hydrogel encapsulation for vocal fold injection.

## 5. Conclusions

Most current tissue engineering in glottic insufficiency focuses on direct injection, biomaterials, cells, or biomolecules independently. Compared to other disease models, the encapsulation of cells and biomolecules in hydrogel offers better synergetic effects. Therefore, this review suggests the potential use of WJMSCs as comparison to ASCs and BMMSCs, due to the ease of obtaining it and its good proliferation. Application of iPSCs should have more studies to yield mature processing techniques and robust outcomes. Encapsulation of VEGF, IGF-1, and EVs in the regeneration of vocal folds should also be elucidate, as that of PRP, bFGF and HGF have been. Based on a review of the literature, the tissue engineering of these stem cells and biomolecules has promising potential in the regeneration of vocal fold muscles and nerves. The recommendation is based on a review of the relevant literature; therefore, additional work and research should be conducted on the suggested cells and biomolecules to provide preliminary evidence for translational application.

## Figures and Tables

**Figure 1 biomedicines-10-03082-f001:**
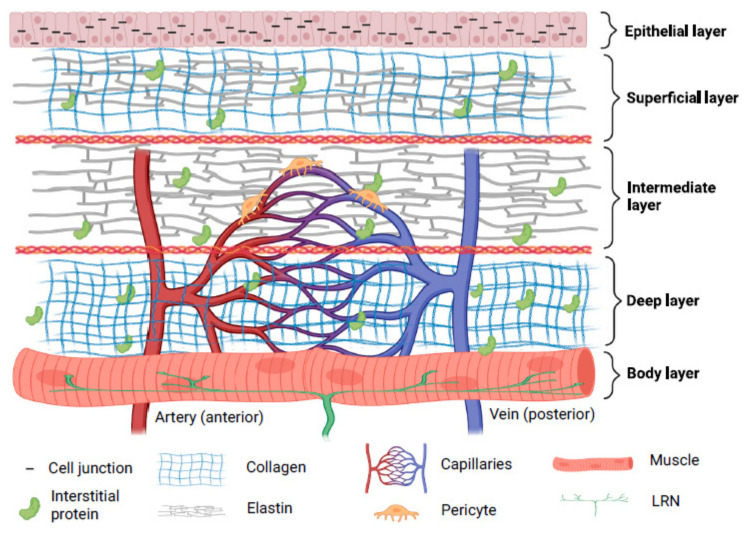
Structure of human vocal fold. Created with BioRender.com (accessed on 24 October 2022).

**Figure 2 biomedicines-10-03082-f002:**
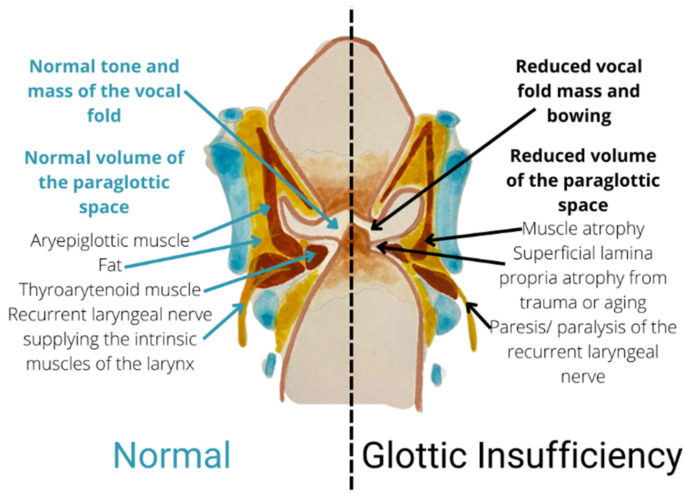
Coronal section showing a comparison between normal condition and glottic insufficiency.

**Figure 3 biomedicines-10-03082-f003:**
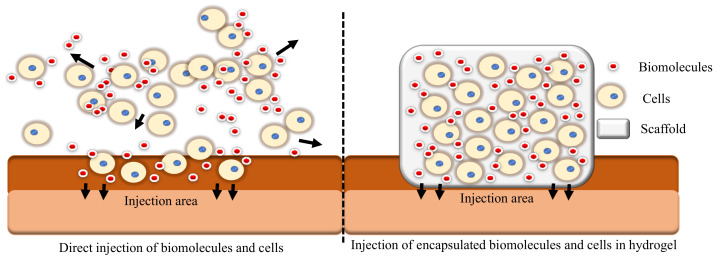
A comparison between direct injection of cells and biomolecules versus injection of cells and biomolecules encapsulated in a hydrogel.

**Figure 4 biomedicines-10-03082-f004:**
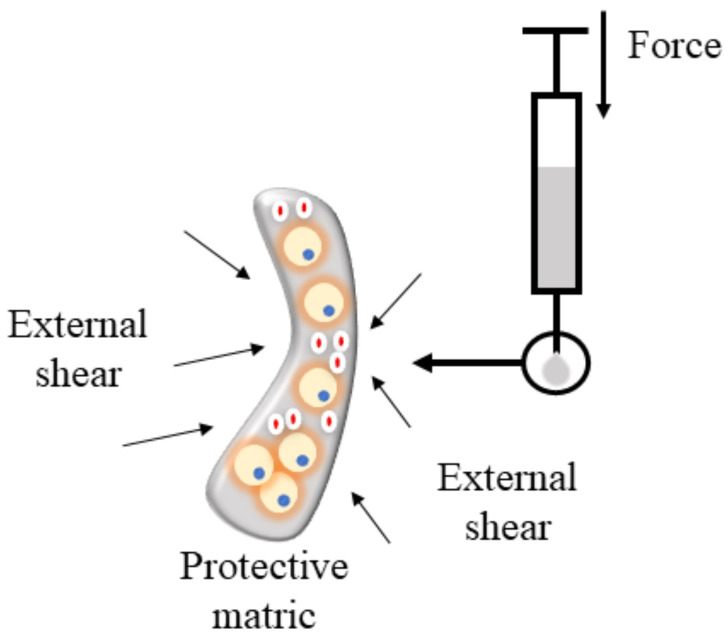
Hydrogel protects cells during injection.

**Figure 5 biomedicines-10-03082-f005:**
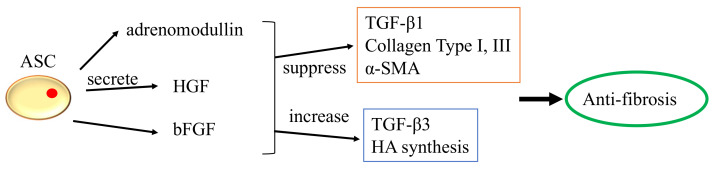
The anti-fibrosis function of ASCs.

**Figure 6 biomedicines-10-03082-f006:**
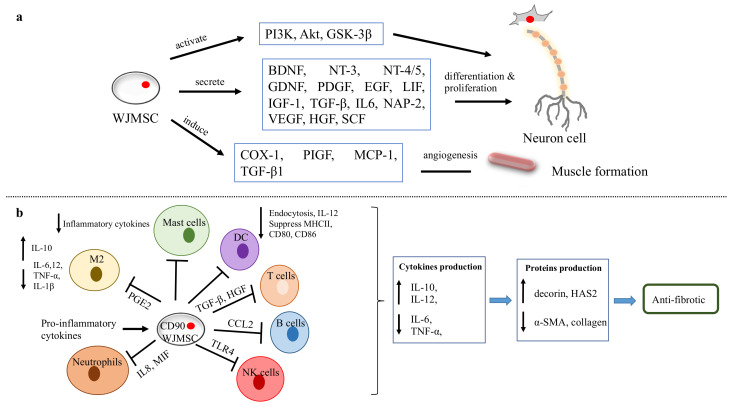
WJMSC functions. (**a**) The function of WJMSCs in neuronal and muscular regeneration. (**b**) Immunomodulatory properties of WJMSCs in preventing fibrosis.

**Figure 7 biomedicines-10-03082-f007:**
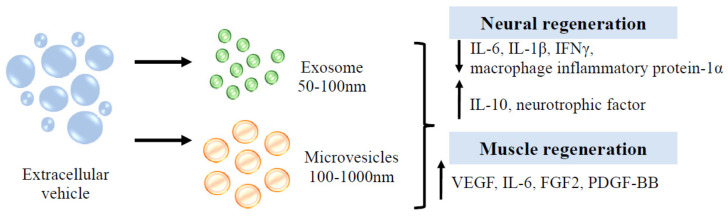
Function of EVs in neuron and muscle regeneration.

**Figure 8 biomedicines-10-03082-f008:**
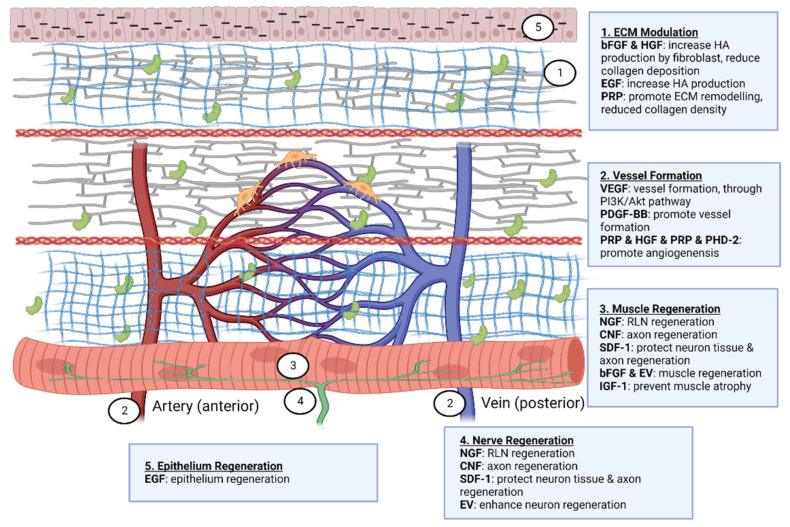
Therapeutic effect of biomolecules on vocal fold regeneration. Created with Biorender.com (accessed on 24 October 2022).

**Table 1 biomedicines-10-03082-t001:** Study outcome and limitation of recent studies on vocal fold injection for regeneration purposes.

No.	Type of Encapsulation	Type of Treatment	Study Design/Study Outcome	ClinicalTrials.gov Number (accessed on 23 October 2022)/Reference
1	Autologous BMMSCs	Encapsulation with hyaluronan gel	Pilot study. Scarred vocal fold is improved.	NCT01981330
		Direct injection	Clinical trial phase 1/2.	NCT04290182
		Limitation: Difficult to source and expand for clinical application.		
2	Autologous ASCs	Encapsulated with injectable collagen scaffold	Clinical trial phase 2.	NCT04164485
		Direct injection	Clinical trial phase 1/2.	NCT02904824
		Direct injection	Improved ECM regeneration in rat model.	[88]
		Direct injection	Clinical trial. Overall voice outcome was improved.	[84,85]
		Limitation: Direct injection yielded short cell retention in providing regenerative effect. It had slower cell proliferation and lesser immunophenotypic indicators than WJMSCs.		
3	bFGF	Direct injection	Clinical trial. Overall voice outcome was improved.	[126,127,128,181]
		Limitation: Single injection was insufficient to obtain satisfactory improvement.		
4	HGF	Direct injection	Clinical trial. Overall voice outcome was improved.	[131]
		Encapsulated with injectable HA/ALG scaffold	HGF in HA/ALG had greater sustained release than direct injection in rabbit model.	[132]
		Direct injection	Re-injection of HGF in rabbit with injured vocal fold reduced collagen expression more significantly.	[133]
		Limitation: Direct injection of HGF had limited retention time for regenerative effect.		
4	PRP & autologous fat	Direct injection	Clinical trial phase 4	NCT04839276
	PRP	Direct injection	N/A	NCT03749863
		Limitation: Applied in short augmentation purpose.		
5	Autologous fibroblast	Direct injection	Clinical trial phase 2	NCT02120781
		Limitation: Difficulty in sourcing available dermal fibroblast for treatment and possible delayed treatment.		
6	Plasmic DNA (pDNA)	Encapsulated in injectable ALG/HA with PCL microspheres	Collagen and HA composition were improved in rabbit with injured vocal fold.	[182]
		Limitation: Complicated components in building suitable hydrogel for pDNA.		

**Table 2 biomedicines-10-03082-t002:** Cells or biomolecules which had potential for hydrogel encapsulation for vocal fold injection (based on literature review).

No.	Type of Encapsulation	Type of Treatment	Study Design/Study Outcome	Reference
1	Human umbilical cord WJMSCs with NGF	Encapsulated in heparinized collagen scaffold	Damaged RLN regenerated in in vivo (rabbit) Scaffold with WJMSCs/NGF had better histomorphological outcome than no WJMSCs/NGF or alone.	[100]
2	iPSCs	Direct injection	iPSCs able to differentiate into skeletal muscle tissue and implanted in thyroarytenoid muscle of rat model. More work needed to ensure safety of iPSCs.	[111]
		Encapsulated in HA hydrogel with EGF	Hydrogel with iPSCs and EGF had less fibrosis in injured vocal fold cells in vitro & rat model.	[108,109]
3	bFGF & HGF	Encapsulated in polycaprolactone (PCL)/pluronic F127 microspheres	Sustained release of bFGF and HGF reduced muscle degeneration and increased muscle regeneration in injured vocal fold of rabbit model.	[183]
4	bFGF	Encapsulated in gelatin microsphere	Scarred formation was reduced in rabbit model. Long term study and inflammation study needed for future study.	[70]
5	VEGF	Encapsulated in microsphere	Improved dental pulp regeneration in mice model.	[136]
6	BDNF & VEGF	Encapsulated in chitosan nanofiber hydrogel	Hydrogel with VEGF provided microenvironment and improved nerve regeneration in rat model.	[139]
7	IGF-1 & MSC	Encapsulated in thermosensitive type 1 collagen	Release of IGF-1 was sustained (2 weeks) and improved MSCs cell proliferation in the hydrogel.	[184]
8	IGF-1 & VEGF	Encapsulated in alginate hydrogel	Release of IGF-1 and VEGF were sustained and improved muscle function in mice and rabbit models.	[185]
9	EVs	Source: WJMSCs Direct injection	Improved nerve regeneration in rat model.	[146]
		Source: BMMSCs Encapsulated in matrix metalloproteinase-2 sensitive self- assembling peptide	Sustained release of EVs in hydrogel and had better outcome of renal function in mice model than direct injection.	[186]

## Data Availability

Data sharing is not applicable to this article as no new data were created or analyzed in this study.

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
