# Peer review of "Tissue Engineering as a Promising Treatment for Glottic Insufficiency: A Review on Biomolecules and Cell-Laden Hydrogel"

_biomedicines, 2022, doi:10.3390/biomedicines10123082_

Round 1

Reviewer 1 Report

The Review article entitled “Tissue Engineering as a Promising Treatment for Glottic Insufficiency: A Review on Biomolecules and Cell-laden Hydrogel” discussed the application of Hydrogel in Tissue regeneration for the Treatment of Glottic Insufficiency. The article has many grammatical and sentence errors, and the language organization needs to be improved. For these reasons, I conclude that the paper should undergo minor revision.

1.      The introduction is good but very general in nature. Authors need to provide more insight into Tissue Engineering and regeneration, various forms of scaffolds, and why Hydrogel is the most preferred for Glottic Insufficiency Treatment.

2.      Revise the Introduction and carefully review the existing and recent Literature.

https://doi.org/10.1016/j.jvoice.2009.10.003

https://doi.org/10.1016/j.biomaterials.2016.08.054

3.      Add a section on the application of bioreactor for Vocal Fold Tissue Engineering

4.      Authors need to explain more about various sources of cells used for Vocal Fold Tissue Engineering with advantages and disadvantages including ESC, ASC and iPSC cells.

https://doi.org/10.1053/j.sempedsurg.2014.04.002

5.      Typographical errors can be avoided. The language and grammar used throughout the manuscript need to be improved. Specific attention needs to be given to this which will improve the standard of the manuscript.

Reviewer 2 Report

1-    There is a high similarity in the manuscript file, and it must be reduced to less than 20% similarity with other publications.

2-    Introduction, line 37, there are 7 references that are too many for a paragraph. Reduce the number of references.

3-    The introduction is well-written but to improve this part, the following related study can be used: https://doi.org/10.1016/j.ijbiomac.2022.01.134

4-    The authors should prepare a Table and summarize the recent studies about injectable hydrogels.

5-    All the figures have low resolution, also a bigger font size should be used in the figures.

6- In section 5.1, the heading can be deleted as everyone knows what tissue engineering is. Instead, it is recommended to summarize recent tissue-engineered studies about Glottic Insufficiency in a Table.

Reviewer 3 Report

This work summarizes the use of tissue engineering for the treatment of glottic insufficiency. My comments are as follows.

1. The numbering of titles should be modified. The current number 2 (2. Structure of Vocal Fold) and number 3 (Etiologies of Glottic Insufficiency can be subtitles to 1. introducation. 

2.  In general, the introduction part of every section is too long, the authors should focus more on the studies done in the literature rather than explanations such as a text book. 

3. Are all figures drawn by the authors?, The writings on the figures are hard to read, should be modified. Figure 5 and 6 can be combined. It is more appropriate to insert the images in the SI to the text also, authors can combine and design the images into figures. I would expect to see figures from the literature studies also. It would be easier to visualize what is done in the literature in this area. 

4. The authors should focus on comparing the literature studies and point out a result.A table containing all the literature examples and contributions, (the treatment technique used, material type, cell type, the effect observed) should be included with ref and year. 

5. The conclusion is insufficient. 

Reviewer 4 Report

This manuscript by Mawaddah and colleagues is a compendium of relevant biomedical engineering related approaches for treatment of glottic insufficiency. Though the descriptions and references are mostly adequate, there are a few key aspects to address:

Issues with grammatical errors across both the abstract and introduction for example in lines 15, 22, 31 and 45 -- this makes the explanations ambiguous to the reader.

Relevant research on modulating natural hydrogel properties for biomedical applications does not include work such as:

Fuchs, S., Shariati, K., & Ma, M. (2020). Specialty tough hydrogels and their biomedical applications. Advanced healthcare materials9(2), 1901396.

Boddupalli, A., Zhu, L., & Bratlie, K. M. (2016). Methods for implant acceptance and wound healing: material selection and implant location modulate macrophage and fibroblast phenotypes. Advanced Healthcare Materials5(20), 2575-2594.

Hinz, B. (2013). Matrix mechanics and regulation of the fibroblast phenotype. Periodontology 200063(1), 14-28.

Please also ensure that there is sufficient DPI for the included figures, as they are pixelating when exported into PDF for review

Round 2

Reviewer 3 Report

Although the revisions are made to an extent by the authors, i believe that the content of the manuscript is not sufficient to be published as a review article in Biomedicines. The introduction and conclusion  still need to be improved.  I suggest the authors to spend more effort on the manuscript and increase the level for publishing. For example, on line 805, the ref 191 is deleted, then it appears on the table. As a whole, the manuscript sounds inattentive. The authors should summarize the literature examples deeper and come to a conslusion, through comparing the literature examples, pointing out the drawbacks, superiorites, etc., and make some suggestions based on these. I suggest an intense investigation of similar review articles in Biomedicines and then going through the manuscript.

Reviewer 4 Report

This revised manuscript by Azman and colleagues has addressed most of the suggestions. Although more attention could have been given to the background of tuning mechanical properties in natural hydrogels, the revisions are mostly appropriate. 

Round 3

Reviewer 3 Report

There are some minor English mistakes, the authors should go over the text. And why do the authors include , MD, Ms, PhD in the authors' name section, i think it is not in the format of the journal. It can be a good idea to remove them.
